# Management of Acquired Hypothalamic Obesity After Childhood-Onset Craniopharyngioma—A Narrative Review

**DOI:** 10.3390/biomedicines13051016

**Published:** 2025-04-22

**Authors:** Hermann L. Müller

**Affiliations:** Department of Pediatrics and Pediatric Hematology/Oncology, University Children’s Hospital, Carl von Ossietzky Universität Oldenburg, Klinikum Oldenburg AöR, 26133 Oldenburg, Germany; hermann.mueller@ymail.com

**Keywords:** craniopharyngioma, obesity, hypothalamus, bariatric surgery, central stimulating agents

## Abstract

**Introduction**: Craniopharyngiomas are rare sellar embryonic malformational tumors of low-grade histological malignancy. Despite high overall survival rates (92%), quality of life is frequently reduced due to adverse late effects caused by hypothalamic obesity. It is well known that morbid hypothalamic obesity is associated with the grade of hypothalamic damage. Accordingly, craniopharyngioma should be considered a paradigmatic disease, reflecting challenges in the diagnosis and treatment of acquired hypothalamic obesity. **Methods**: A narrative review was performed after searching the MEDLINE/PubMed, Embase, and Web of Science databases for initial identifying articles. The search terms childhood-onset craniopharyngioma and hypothalamic obesity were used. **Results**: Despite the availability of promising therapeutic approaches, such as medication with central stimulating agents, antidiabetic drugs, glucagon-like peptide 1 (GLP1) receptor agonists, and Setmelanotide, it must be emphasized that there is currently no pharmaceutical treatment for hypothalamic obesity in craniopharyngioma proven to be effective in randomized controlled trials. For Setmelanotide, a prospective blinded randomized trial over a 12-month treatment period is ongoing. Bariatric interventions are effective, but non-reversible procedures such as bypass operations are controversial in the pediatric age group due to legal and ethical concerns. Recently, a treatment algorithm was introduced to improve the management of hypothalamic syndrome/obesity by offering more personalized treatment. Decisions on treatment strategies focusing on the preservation of visual, neuroendocrine, and hypothalamic integrity should be made by experienced multidisciplinary teams. **Conclusions**: Treatment approaches for hypothalamic obesity are limited. Further research on novel treatment approaches for hypothalamic obesity is warranted to improve the quality of life after childhood-onset craniopharyngioma.

## 1. Introduction

Childhood-onset craniopharyngiomas (CPs) are embryonic tumors of low-grade histological malignancy, originating from the sellar and parasellar regions of the skull base. The WHO 2021 classification of central nervous system tumors defines adamantinomatous and papillary CPs, for the first time, as two distinct tumor entities. Adamantinomatous CP (ACP) is mainly diagnosed in children and adolescents (<18 years), whereas papillary CP (PCP) is diagnosed almost exclusively in adults. The median age at the time of diagnosis ranges from 5 to 9 years for ACP and from 55 to 69 years for PCP. The clinical picture at the time of diagnosis is often characterized by non-specific manifestations of intracranial pressure. The diagnosis of CP is often made late, sometimes years after the initial appearance of symptoms. Among adult-onset CP, the diagnosis is made mainly based on visual symptoms, as well as longstanding endocrine insufficiencies. The incidence of CP is 0.5–2.0 new CP cases per million persons per year [1] (Figure 1). Despite a high overall survival rate (92%), the quality of life after CP is frequently reduced due to adverse late effects caused by morbid obesity [2]. It is well known that the grade of hypothalamic obesity in affected CP patients is associated with the degree and extent of hypothalamic lesions [3]. Therefore, CP should be considered a paradigmatic, mostly chronic disease, reflecting problems and challenges in the diagnosis, treatment, and follow-up care of patients with acquired hypothalamic obesity [4]. This narrative review aims to present an overview of the current knowledge about hypothalamic obesity and its treatment, as well as future directions of research in this area.

After searching PubMed/MEDLINE, Web of Science, and Embase for the initial identification of articles, a narrative review was performed. Craniopharyngioma and hypothalamic obesity were used as search terms. Selected English-language papers published between 1995 and 2024 were included in this narrative review.

## 2. Anatomical Location of Childhood-Onset Craniopharyngioma Close to the Hypothalamus

Due to the location of CPs, the proximity to the hypothalamus and pituitary gland, and the risks and challenges of treatment, CP patients frequently present with hormonal dysfunction and develop (neuro)-endocrine complications such as stunted growth, hypocortisolism, hypothyroidism, arginine vasopressin deficiency, and hypogonadism. Additionally, some CPs are located close to the optic nerves and optic chiasm and might, therefore, cause visual impairment or even blindness through mechanical compression [2]. The hypothalamus includes different nuclei, which have various functions (Figure 2).

The hypothalamus controls information used to regulate metabolic homeostasis, including information from the environment such as temperature, light, pain, and smell; internal information from the body such as glucose concentration, blood pressure, osmolality, and hormonal secretion through the hypothalamic–pituitary–adrenal axes; and information from the central nervous system [7]. Accordingly, the hypothalamus regulates growth, satiety and thirst, circadian rhythm, body temperature, circulatory function, stress response, and sexual and reproductive functions [7]. The hypothalamus is bordered anteriorly by the optic chiasm and caudally by the thalamus [7]. Medially, it forms part of the wall and floor of the third ventricle and merges into the infundibulum of the pituitary gland dorsally in the area of median eminence at the tuber cinereum [7]. In the region of the arcuate nucleus and the paraventricular nucleus, releasing hormones are produced and released into the portal vein system within the median eminence (Figure 2) [7]. This network connects the hypothalamus and the pituitary gland. Additionally, magnocellular neurons are located within the paraventricular nucleus and the supraoptic nucleus, projecting their axons into the neurohypophysis [7]. These neurons influence the secretion of vasopressin and oxytocin into the peripheral vascular system [8]. Neurons in the paraventricular nucleus, ventromedial nucleus, dorsomedial nucleus, and lateral hypothalamic area control food intake [7]. The median preoptic nucleus, subfornical organ, and organum vasculosum form the thirst center in the lamina terminalis [7].

## 3. Diagnosis, Treatment, and Follow-Up Care of Patients with Childhood-Onset Craniopharyngioma

Children diagnosed with CP have a median age of 5 to 12 years old and present with varied initial symptoms and complaints. Headache, stunted growth, polyuria/polydipsia, and visual impairment can be the first signs of a CP during childhood and adolescence [9,10]. When CP is suspected, patients should ideally undergo comprehensive whole-brain magnetic resonance imaging (MRI), assessing the sellar and suprasellar area (Figure 3) via 3D high-resolution coronal, axial, and sagittal T2-weighted images and volumetric T1-weighted sequences. For initial MRI scans of CPs, using an intravenous contrast agent is recommended. To guide diagnosis, MRI should be supplemented with computed tomography (CT) scans (limited sellar scans sparing the lenses, without an intravenous contrast agent) to visualize calcifications. After neurosurgical intervention, an MRI scan within 4 weeks serves as the initial postoperative imaging method for assessing the degree of surgical resection [11]. Typically, CP is located at a single central nervous system location, being unifocal and often having solid and cystic tumor components [12]. Rare cases of ectopic CP have been reported [13].

Morphologically, different grading systems of tumor location for hypothalamic involvement have been introduced in the literature. Puget et al. [14] developed a radiological scoring system and described three levels of preoperative hypothalamic involvement: grade 0 refers to no hypothalamic involvement, grade 1 to a tumor abutting or displacing the hypothalamus, and grade 2 to hypothalamic involvement, where the hypothalamus is no longer identifiable. Postoperatively, grade 0 refers to no hypothalamic damage, grade 1 to negligible hypothalamic damage or a residual tumor displacing the hypothalamus, and grade 2 to significant hypothalamic damage, where the floor of the third ventricle is no more identifiable [14]. Müller et al. introduced a neuroradiological grading system, where hypothalamic involvement was assessed preoperatively and hypothalamic lesions were assessed postoperatively [3,15]. Regarding preoperative hypothalamic involvement, grade 0, according to the Müller grading system, refers to no involvement of the hypothalamus by the tumor, grade 1 refers to involvement of the anterior hypothalamic area excluding the mammillary bodies and posterior hypothalamic areas beyond the mammillary bodies, and grade 2 refers to an anterior hypothalamic involvement including mammillary bodies and posterior hypothalamic areas beyond mammillary bodies [3,15]. Postoperatively, for a hypothalamic lesion, grade 0 refers to no hypothalamic lesion being present, grade 1 refers to anterior damage without a lesion affecting the mammillary bodies and the posterior hypothalamic parts, and a grade 2 lesion includes anterior parts, mammillary bodies, and posterior hypothalamic areas [3,15].

Genetically, childhood-onset CPs mostly exhibit the adamantinomatous type marked by a CTNNB1 (gene encoding β-catenin) mutation, activating the WNT pathway [16]. Tumor development is linked to inflammation, particularly increased levels of cytokines (e.g., Interleukin 6) [17].

Treatment usually includes surgical resection of the tumor, with or without subsequent radiotherapy of the residual tumor after incomplete resection. The treatment decision requires an experienced multidisciplinary team to evaluate the benefits and risks of treatment for each case individually. The first, often controversial, decision regarding the treatment of CP is whether a gross total resection can be achieved safely without surgically damaging critical neighboring cerebral structures. A consensus is that surgical CP resection should always aim to secure the integrity of the hypothalamus, visual structures, and critical skull base vessels. Subsequently, surgical interventions should also aim to reduce tumor volume while leaving parts of the tumor in critical locations unresected. Afterward, irradiation is frequently applied for treating residual CPs, though the timing of radiotherapy after surgery varies. Since CPs can also consist of cystic parts, cyst fenestrations are also applied to reduce cystic and intracranial pressure. Depending on the location of the tumor and the experience of the surgeon, a trans-sphenoidal or transcranial approach is used. Given the different characteristics of CPs, there is no “one-size-fits-all” approach for neurosurgical treatment and radiotherapy [18]. Risk-adapted decisions using individualized treatment strategies are required.

Proton beam therapy, as a modern, highly conformal, intensity-modulated irradiation technique, has been used more frequently in the past decade due to its increased practicality [19]. According to a US phase 2 clinical trial, proton beam therapy did not enhance survival rates in pediatric and adolescent patients with CP compared to historical cases, and severe complication rates remained consistent [20]. Nonetheless, cognitive outcomes were improved after proton beam therapy compared to photon-based therapy [20]. Children and adolescents treated for CP via subtotal resection and postoperative proton beam therapy have high tumor control rates and a low incidence of severe complications [20]. However, clinically relevant differences in quality of life, cognitive outcomes, and hypothalamic syndrome are still topics of debate. Furthermore, evaluating long-term outcomes such as the incidence of second malignant neoplasms and vasculopathies requires longer observational times [19].

There have been recent reports on deep brain stimulation (DBS) for treating hypothalamic obesity, which show that it has the potential to improve the quality of life in patients suffering from acquired hypothalamic obesity [21,22].

After initial surgery, MRI scans should be performed approximately every 3 months during the first postoperative year for longitudinal follow-up on the disease response [11]. Clinical manifestations present before and after treatment for CP include visual dysfunction, which is present in approximately 50% of children at CP diagnosis [23]. The post-surgical recovery of initial visual deficits has been observed in several cohorts, varying between 33 and 47% [24,25,26]. During follow-up care, hyperphagia [27], temperature dysregulation, sleep problems [28,29], arginine vasopressin deficiency (AVD), and hypopituitarism are complex symptoms associated with CP.

## 4. Survival After Childhood-Onset Craniopharyngioma

Thirty-year survival rates of childhood-onset CP patients are favorable (up to 80% overall survival) but frequently interrupted by numerous relapses and interventions, which lead to considerable long-term morbidity [30]. Survival of childhood-onset CP is also influenced by (neuro)endocrine deficits. Depending on tumor location and treatment-associated lesions, different hypothalamic–pituitary axes are affected. The endocrinopathies include deficiencies of thyroid-stimulating hormone (TSH), growth hormone, luteinizing hormone (LH)/follicle-stimulating hormone (FSH), adrenocorticotropic hormone (ACTH), and AVD. In one of the first large analyses on patients with CP, de Vile et al. described panhypopituitarism in 75% of the investigated samples [31].

In an international survey conducted by the Dutch Pituitary Gland patient group, patients with hypothalamic and pituitary deficiencies were asked which symptoms affected them the most during their daily lives [32]. Hypothalamic obesity and fatigue appeared to be the biggest problems for them [32]. Treating metabolic syndrome (arterial hypertension, lipid disorders, and glucose intolerance) as a consequence of hypothalamic obesity is frequently part of long-term follow-up.

Furthermore, long-term sequelae of radiotherapy can impact survival and quality of survival after CP. Besides secondary malignancies, which can appear years after irradiation exposure, vascular morbidity might be increased in patients after radiotherapy for treating CP [33,34,35].

## 5. Hypothalamic Syndrome

Hypothalamic syndrome is an umbrella term that describes clinical manifestations occurring in patients with hypothalamic lesions or deficiencies [4]. Patients with CP, as well as patients with other diseases such as suprasellar brain tumors (pilocytic glioma and germ cell tumors) or genetic diseases (Prader–Willi syndrome and Bardet–Biedl syndrome), with septo-optic dysplasia (SOD), and after traumatic brain injury (TBI), frequently present with symptoms of hypothalamic syndrome. Hypothalamic syndrome is not synonymous with hypothalamic obesity. Hypothalamic syndrome consists of multiple clinical characteristics and might be clinically different for each patient [36]. Van Santen et al. have described five clinical domains for diagnosing hypothalamic syndrome: eating behavior disorders, dysregulation of circadian rhythms and sleep, behavioral disorders, temperature regulation disorders, and hormonal deficiencies [37,38]. The below list explores these issues in more detail:Eating behavior disorders include hypo- and hyperphagia [27], which are characterized by severe hunger, binge eating, increased weight, and obesity or underweight up to cachexia, which is more likely to occur at the time of CP diagnosis and before surgical intervention [37].Behavioral disorders include compulsive symptoms, obsessive behavior, rage, and hoarding [39].Dysregulation of circadian rhythms and sleep issues include sleep apnea, hypersomnia, increased daytime sleepiness, insomnia, and fatigue [37]. Fatigue is a multidimensional symptom present if exhaustion or tiredness cannot be related to a certain activity [40]. Patients with hypothalamic syndrome might report fatigue symptoms; however, fatigue is also a recognized late effect after other pediatric cancers and can be present in patients without sleep disorders (e.g., sleep apnea and narcolepsy) [41]. Accordingly, it can also be associated with CP treatment (e.g., irradiation).Dysregulation of temperature issues include hyper- and hypothermia and discomfort with cold or warm extremities and face [37,42].Disorders of pubertal development such as central precocious puberty or delayed pubertal development and deficiencies of growth hormone, TSH, ACTH, and LH/FSH are endocrine dysfunctions described for hypothalamic syndrome [37]. Arginine vasopressin deficiency (AVD, formerly central diabetes insipidus) with or without adequate thirst feeling can be included [37].

Patients with hypothalamic syndrome have an increased risk of cardiovascular complications [35] and morbidity related to metabolic disorders [4]. The above-mentioned clinical manifestations of hypothalamic syndrome significantly impact the health-related quality of life of affected patients (Table 1).

## 6. Pharmacological Treatment of Patients with Acquired Hypothalamic Obesity

Intractable weight gain after tumor- and/or treatment-associated hypothalamic damage is a known course of CP disease leading to morbid hypothalamic obesity. Severe hypothalamic obesity occurs in approximately 40 to 60% of patients with hypothalamic injury-related long-term sequelae after CP [18]. Physiologically, weight and appetite are regulated through orexigenic (e.g., neuropeptide Y, ghrelin, and orexin A and B) and anorexigenic (e.g., insulin, leptin, adiponectin, and brain-derived neurotrophic factor (BDNF)) pathways [54]. These pathways can be disturbed due to the tumor- and/or treatment-related damage of specific hypothalamic nuclei such as the nucleus arcuatus, ventromedial nucleus, and nucleus tractus solitarius [54]. Resistance to leptin and insulin are clinical consequences [55]. Furthermore, eating behavior disorders including imbalances of satiety and appetite sensation can be observed as adverse effects [38]. Due to increased vagal activation, CP patients with hypothalamic obesity present with parasympathetic dominance of their autonomic nervous system [56]. Lower levels of total energy expenditure are frequently observed due to decreased sympathetic activity. Parasympathetic dominance results in long-term fat accumulation and hyperinsulinemia [38].

Once patients have reached morbid levels of obesity, a reduction in obesity can hardly be achieved [38]. To date, no “one-size-fits-all” standard treatment of hypothalamic obesity exists in these clinically challenging cases [54]. Current strategies include preventing or minimizing hypothalamic lesions via hypothalamus-sparing surgeries, novel radio-oncological techniques (e.g., proton beam therapy), and novel pharmacological agents for treating hypothalamic obesity [57].

Morbid obesity due to hypothalamic damage is mostly non-responsive to conventional lifestyle modifications such as physical exercise or dietary interventions [58]. Central stimulating pharmaceutical agents such as methylphenidate, dextroamphetamine, and tesomet increase the metabolic rate and suppress appetite [59,60]. A systematic review by van Iersel et al. observed weight reduction or stabilization in 89% of patients with hypothalamic obesity [38]. Central stimulating agents may also have beneficial effects on fatigue, increased daytime sleepiness, hyperphagia, and psychosocial disorders [38].

Pharmacological agents with antidiabetic effects (i.e., glucagon-like peptide (GLP)-1 receptor agonists, metformin, diazoxide, fenofibrate, and pioglitazone) can reduce weight gain and improve insulin sensitivity. Metformin seems to stabilize and reduce weight development with the least side effects. However, the long-term effects of metformin medication on hypothalamic obesity are unknown [61]. The beneficial effects of GLP-1 receptor agonist treatment on weight development in adult patients with hypothalamic obesity have been discussed but are controversial. The benefits of decreased food intake and improved glycemic control have been reported [62,63]. Semaglutide has been approved by the FDA for obesity treatment in adolescents, and a median BMI decrease of 17% has been reported. It has also been speculated that semaglutide could have beneficial effects on weight development in patients suffering from acquired hypothalamic obesity [64,65,66]. Early studies suggested that GLP-1 receptor agonists might increase the risk of thyroid cancer [67]. A large-scale Scandinavian study found no significant increase in thyroid cancer risk in those taking GLP-1 receptor agonists compared with those on other treatments for type 2 diabetes [68].

The neuropeptide oxytocin suppresses appetite and hunger sensation. Changes in the salivary concentrations of oxytocin triggered by food intake and exercise seemed to be associated with eating behavior and BMI in CP patients. These observations led to speculation that oxytocin treatment might have a beneficial effect on hypothalamic obesity [69,70]. However, there are still unanswered questions regarding pathomechanisms of action and dosage of oxytocin in treating patients with CP and hypothalamic syndrome.

Setmelanotide, a melanocortin-4 receptor (MC4R) agonist originally developed for treating congenital hypothalamic obesity caused by pro-opiomelanocortin (POMC) or leptin receptor (LEPR) deficiency [71], is known to stimulate remaining (hypothalamic) MC4R neurons [55]. Disruptions of the MC4R pathway lead to hyperphagia and consecutive hypothalamic obesity. In a recent phase 2 study in adults and children, Roth et al. observed an impressive reduction in BMI (mean BMI reduction of 15% ± 10%), clinically relevant mitigating effects on hyperphagia, and beneficial effects on quality of life [72]. Currently, the effects of Setmelanotide on weight development and hyperphagia in patients with hypothalamic obesity are being analyzed through an ongoing prospective blinded randomized trial over a 12-month treatment period. Setmelanotide might provide a new and promising therapeutic perspective on hypothalamic obesity [73] (Table 2).

## 7. Bariatric Surgery

Bariatric interventions demonstrated notable effectiveness in weight loss and safety in patients with CP [84,85,90] (Table 1). In a 2013 meta-analysis, Bretault et al. [84] reviewed 21 cases and observed significant weight reduction at 6 months and 12 months after bariatric intervention (−20.9% weight loss at 6 months; −15.1% weight loss at 12 months). The greatest weight reduction was achieved after the Roux-en-Y gastric bypass (RYGB). In pediatric patients, the application of non-reversible bariatric techniques such as bypass interventions is ethically and legally controversial. The decision for bariatric treatment should be made by multidisciplinary teams in the context of national/international trials [90]. The Endocrine Society has published a Clinical Practice Guideline for Treating Pediatric Obesity [91], suggesting that only adolescents with severe obesity (BMI > 40 kg/m^2^ or BMI > 35 kg/m^2^ plus comorbidities), Tanner > 3 pubertal stage, and final or near-final adult height should be considered for bariatric surgery. Postoperative adverse side effects after bariatric surgical interventions include anemia due to iron deficiency, gastrointestinal complications such as diarrhea, dumping syndrome due to RYGB, vitamin D and folic acid deficiency, impaired resorption of oral vasopressin medication following sleeve gastrectomy, and dysphagia and vomiting due to laparoscopic gastric banding (LAGB) [90]. As patients with severe hypothalamic obesity frequently suffer from varied neuroendocrine dysfunctions, the specific benefits of bariatric interventions on hypothalamic obesity are not well documented. Patients with hypothalamic obesity should be evaluated individually to decide whether they are candidates for bariatric intervention and which bariatric procedure is most appropriate.

## 8. Conclusions

Acquired hypothalamic obesity causes severe morbidity and impairment of quality of life in survivors of CP. Treatment for hypothalamic obesity after CP has thus far been rather disappointing. Accordingly, hypothalamus-sparing treatment strategies are currently the most effective therapeutic options for preventing hypothalamic syndrome and acquired hypothalamic obesity. Standardizing the diagnostic criteria for hypothalamic obesity/syndrome and surgical and radio-oncological interventions could help to improve treatment quality. In adult-onset CP, targeted therapy of *BRAF V600E* mutation-positive CP is promising. Similar approaches in pediatric CP could help to treat and reverse hypothalamic tumor infiltration as a major pathogenic component in acquired hypothalamic obesity. Novel pharmaceutical agents for treating hyperphagia and hypothalamic obesity such as Setmelanotide have shown promising results. Further developing new agents and testing them in randomized trials could help to provide promising new perspectives for treating patients with hypothalamic obesity. Decisions on treatments should be made by experienced multidisciplinary teams. Due to the rareness of the disease, collaboration in international trials and registries is necessary.

## Figures and Tables

**Figure 1 biomedicines-13-01016-f001:**
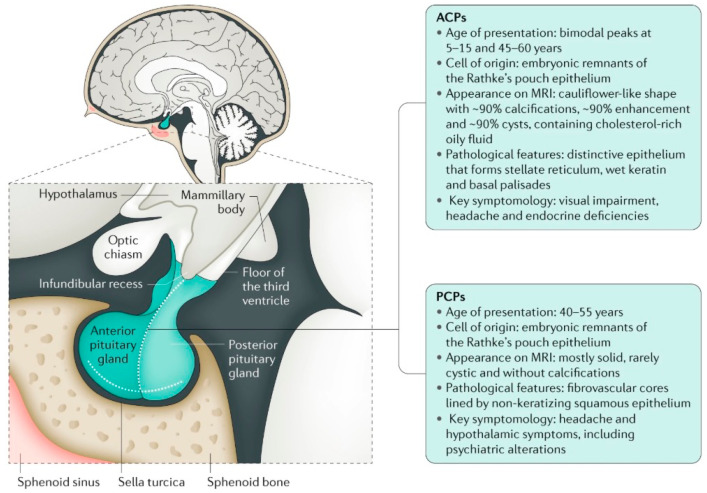
Features of adamantinomatous craniopharyngioma (ACP) and papillary craniopharyngioma (PCP). Reproduced from Müller, H.L. et al. (2019) [5] Craniopharyngioma. *Nat. Rev. Dis. Primers* with the kind permission of Springer Nature.

**Figure 2 biomedicines-13-01016-f002:**
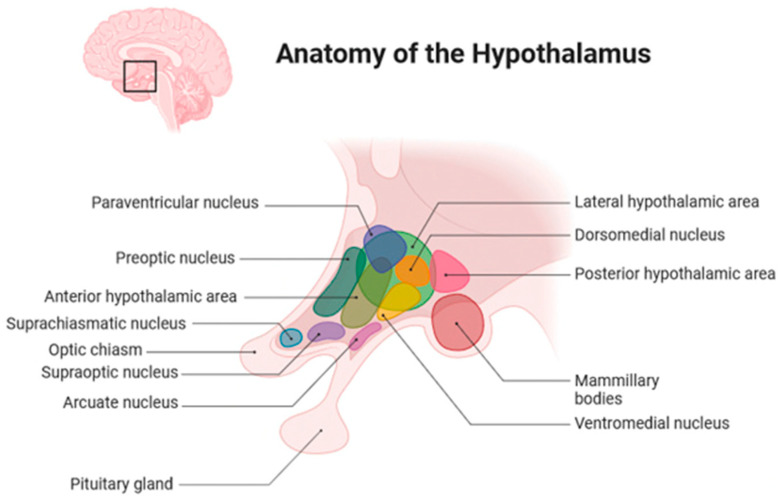
Anatomical scheme of hypothalamic nuclei, including different hypothalamus [created with a graphical software (BioRender; www.biorender.com/, Toronto, CA, USA)] [6].

**Figure 3 biomedicines-13-01016-f003:**
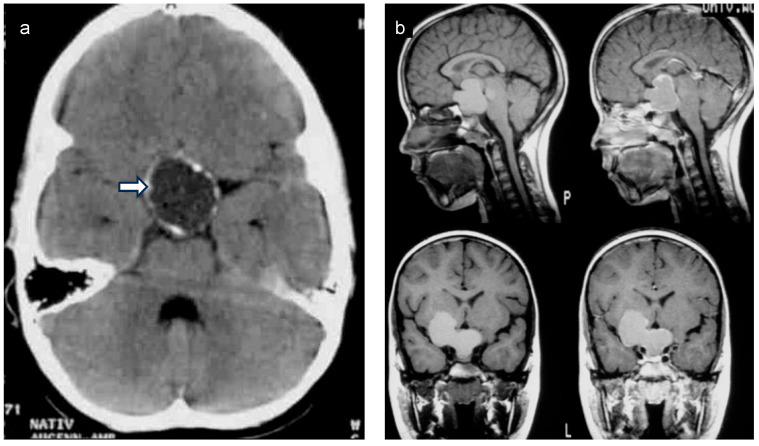
(**a**) Native coronal computed tomography (CT) of the skull of a patient with childhood-onset craniopharyngioma, depicting calcifications in the cyst membrane of a craniopharyngioma (arrow). (**b**) Sagittal and frontal magnetic resonance imaging (MRI) of a patient with childhood-onset craniopharyngioma. Both patients were recruited in the KRANIOPHARYNGEOM 2007 trial.

**Table 1 biomedicines-13-01016-t001:** Health-related quality of life after childhood-onset craniopharyngioma. Selected publications (2019–2024).

Diagnosis	Pat. No.	Age at Craniopharyngioma Diagnosis (Years)	Follow-Up Interval (Years)	Treatment	Quality of Life/Outcome	Authors/Year of Publication
CO CP	48	GTR: Median 6.4 (range: 2.2–16.8) PR + RT: Median 8.5 (range: 3.8–16.4)	10 years	21 GTR, 22 PR + RT	No differences in the trajectory of intellectual functioning or QoL scale scores between the two groups (GTR vs. PR + RT).	Aldave et al., 2023 [43]
CO CP from a lower-middle-income country	29	Mean age 13.5 ± 4.2 SD	Mean FUP: 4.4 ± 2.2 SD	15 GTR 11 Debulking 3 Reservoir and biopsy	PedsQL: GTR 56.6 ± 7.12 Debulking: 93.8 ± 3.37 Biopsy: 83.3 ± 5.69	Baqai et al., 2024 [44]
CO CP with presurgical grade 2 HI [3]	109	Median 9.5 (range: 1.3–17.9)	Mean 6.1 (range: 3.0–10.2)	Surgery leading to 23 grade 0 HL 29 grade 1 HL 57 grade 2 HL	Worse PEDQOL for grade 3 patients in terms of physical, social, and emotional functionality when compared with HL grade 0 and 1.	Bogusz et al., 2019 [45]
Caregivers of CO CP patients	82	Mean age 9.3 ± 4.5 SD		52.4% GTR	Survivor poly-symptomatology predicted caregiver burden. The study separated hyperphagia and obesity and identified hyperphagia and other hypothalamic dysfunction symptoms as understudied issues.	Kayadjanian et al., 2023 [46]
Hypothalamic dysfunction + CO CP	290	n.a.	n. a.	n. a.	Worldwide online survey: Obesity (51%) and fatigue (48%). Needs for improvement in the domains of obesity, fatigue, and lifestyle.	Van Roessel et al., 2020 [32]
CO CP	119	Median 12 (range: 2–17)	Mean 10 (range: 1–39)	CR in 34 (29%) 6 HL Garde 0 23 HL grade 1 55 HL grade 2	QoL (EORTC QLQ-C30) was negatively correlated with daytime sleepiness (ESS), the highest ESS in patients with HL grade 2.	Mann-Markutzyk et al., 2025 [47]
CO CP	131	Median 9.7 (range: 1.3–17.6)	3 years	Complete Res.: 21 (18%) Incomplete Res.: 94 (82%)	Grade 2 HI, grade 2 HL, and complete surgical resection were associated with low QoL.	Eveslage et al., 2019 [48]
Caregivers of CO CP patients	106	<18 years	n. a.	48 RT 134 surgical interventions	Online survey: reduced social functioning.	Craven et al., 2022 [49]
CO CP	92	Mean age 10.5 ± 4.0 SD	n. a.	Proton beam therapy after surgical intervention	Fatigue, QoL, and brain tumor symptoms improved over time during proton beam therapy.	Mandrell et al., 2024 [50]
CO CP	78	Mean age 10.8 ± 3.11 SD	n. a.	56 surgical resections 16 catheter implantation	Poorer parent-reported QoL; AVD directly predicted greater global executive functioning impairment.	Niel et al., 2021 [51]
CO CP and parents/caregivers	120	Median 10.0 (range: 1.3 –16.8)	3 years	25 complete resection 95 incomplete resection 61 RT	Reduced autonomy was found three years after diagnosis in self-assessments and parental assessments of QoL (PedQol).	Sowithayasakul et al., 2023 [52]
Irradiated CO CP	99	Median 9.5 (range: 1.6–17.9)	Median 6.4 (range: 0.9–14.7)	64 proton beam therapy 35 photon-based RT	No significant difference between PBT and RT in terms of QoL (PedQol), functional capacity (FMH), and body mass index.	Friedrich et al., 2023 [19]
CO CP	87	Mean 7.39 ± 3.67 SD	Median 6.54 (IQR: 3.11–10.69)	25% complete resection 44% incomplete resection 30% cyst drainage 46% RT	BMI at dgn and grade of HL were associated with hypothalamic obesity.	Van Schaik et al., 2023 [53]

Abbreviations: CO CP, childhood-onset craniopharyngioma; QoL, quality of life; GTR, gross total resection; HI, presurgical hypothalamic involvement; HL, surgical hypothalamic lesions; PR, partial resection; RT, radiotherapy; PBT, proton beam therapy; dgn, diagnosis; AVD, arginine vasopressin deficiency; n. a., data not available; IQR, interquartile range; SD, standard deviation; ESS, Epworth Sleepiness Scale; pat, patients; and FMH, functional ability scale Münster Heidelberg.

**Table 2 biomedicines-13-01016-t002:** Therapeutic approaches for treating acquired hypothalamic obesity and effects on weight development/obesity.

	Intervention	#	Age (Years)	BMI/Weight at Intervention	BMI/Weight Change During/After Intervention	Authors
Pharmacological agent	Dextroamphetamine	19	12.3 ± 4.0	BMI 3.58 ± 0.85 SD	ΔBMI SDS −0.14	van Schaik et al. [74]
	Dextroamphetamine	7	0.5, 11.1, 11.8, 12.5, 14.7, 14.8, 21.0	BMI +3.17 ± 0.9 SD Range: +1.9 to +4.4 SD	Mean BMI SDS decelerated to −0.18 ± 0.12/year during the 1st year of treatment and stabilized at +0.05 ± 0.32/year during the 2nd year of treatment.	Denzer et al. [60]
	Diazoxide/metformin	9	15.4 ± 2.9	BMI +1.8–+2.96 SD	ΔBMI −0.3 ± 2.3 kg/m^2^	Hamilton et al. [75]
	Octreotide (RCT)	10	13.8 ± 1.2	BMI 37.1 ± 1.3 kg/m^2^	ΔBMI −0.2 ± 0.2 kg/m^2^ (vs. placebo +2.2 ± 0.5 kg/m^2^)	Lustig et al. [76]
	Semaglutide	26	52 (18–65)	BMI 38 (28–58) kg/m^2^.	Mean TWL 13.4 kg (95% CI 10.3–16.5 kg)	Svendstrup et al. [77]
	Semaglutide	4	22, 44, 57, 69	BMI 48.0 (35.0–55.5) kg/m^2^	ΔBMI 7.9 BMI (6.7–10.1); weight loss of 17.0% (11.3–22.4%)	Gjersdal et al. [65]
	Exenatide/liraglutide	9	46 (22–49)	BMI 37.6 ± 7.2 kg/m^2^	Exenatide: ΔBMI −6.1 to −2.8 kg/m^2^; liraglutide: Δweight −22 to −9 kg	Zoicas et al. [78]
	Exenatide	8	27.5 ± 7.8	BMI 47.5 ± 10.8 kg/m^2^	Mean Δweight −1.4 kg	Lomenick et al. [79]
	Tesomet (tesofensine and metoprolol)	18	45.4 ± 13.3	BMI 37.3 ± 5.6 kg/m^2^	Δweight: −6.3% (tesomet −6.6% vs. placebo −0.3%)	Huynh et al. [59]
	Setmelanotide	18	15.0 ± 5.3	BMI 38·0 ± 6·5 kg/m^2^	ΔBMI: −15% (SDS 10%) after 4 mo; extension 12 mo (12 patients): −26% (12 SDS)	Roth et al. [72]
Lifestyle modification	Regular visits to a comprehensive care clinic	39	13.4 (4.3–18.2)	BMI 1.93 (0–3.2) SD	Median ΔBMI rate +4.5 kg/m^2^/y (−17.8 to +8.4); median ΔBMI SDS rate 0.0/y (−5.2 to +0.5)	Rakhshani et al. [58]
Bariatric surgery	SG (n = 3); RYGB (n = 5)	8	33.4 ± 13.6	BMI 43.3 ± 4.1 kg/m^2^	SG (n = 3): mean Δweight −10%; RYGB (n = 5): mean Δweight −25%	Wijnen et al. [80]
	RYGB (n = 12), SG (n = 4)	16	26 ± 12	BMI 46 ± 8 kg/m^2^	Mean Δweight: −22% after 5 years	van Santen et al. [81]
	SG (39%), RYGB (61%)	23	35 (25–43)	BMI 44.2 (40.7–51.0) kg/m^2^	ΔTWL (%) − 39.0% (14.0; 53.3)	Faucher et al. [82]
	SG (n = 2), RYGB (n = 3)	5	38 (27–47)	BMI 41.3 (37.9–46.3) kg/m^2^	ΔTWL (%) −14.7% (23.7; 5.8)	Garrez et al. [83]
	LAGB (*n* = 6); SG (*n* = 8); RYGB (*n* = 6); BPD (*n* = 1)	21	24 (12–54)	BMI 49.6 kg/m^2^	TWL (%) LAGB: 10.5%; SG: 20.7%; RYGB: 20.2%; BPD: 24.8%	Bretault et al. [84]
	LAGB (n = 4)	4	13, 17, 21, 23	BMI +7.3 – +12.3 SD	ΔBMI +1.7 to +8.7 kg/m^2^	Müller et al. [85]
	LAGB (n = 6); SG (n = 4); RYGB (n = 2)	9	17 (12–30)	BMI 44.7 (40.2–61.6) kg/m^2^	LAGB: no change; SG: no change; RYGB: mean Δweight −30%	Weismann et al. [86]
	SG	3	21, 22, 24	BMI 49.2 (41.6–58.1) kg/m^2^	Mean ΔBMI −13.9 kg/m^2^; Δweight −17.6%, −25.0%, −41.1%	Trotta et al. [87]
	SG (n = 2); RYGB (n = 2)	4	24, 30, 43, 51	BMI 37.6, 37.7, 43.7, 51.0 kg/m^2^	ΔBMI: SG −10, −3.6; RYGB: −6.2, +11.3 kg/m^2^	Gatta et al. [88]
Vagotomy	Truncal vagotomy	1	19	BMI 43.0 kg/m^2^	Δweight −7.0 kg	Smit et al. [89]
DBS	Nucleus accumbens DBS	1	19	BMI 52.9 kg/m^2^	ΔBMI −5.2 kg/m^2^	Harat et al. [21]

Abbreviations: #, cohort size/patient number; TWL, total weight loss; BMI, body mass index; SDS, standard deviation score; LAGB, laparoscopic adjusted gastric banding; RYGB, Roux-en-Y gastric bypass; SG, sleeve gastrectomy; DBS, deep brain stimulation; RCT, randomized controlled trial; year, yr; and n. a., data not available.

## Data Availability

The datasets generated and/or analyzed during the current study are available from the author upon reasonable request.

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
