# Peer review of "Management of Acquired Hypothalamic Obesity After Childhood-Onset Craniopharyngioma—A Narrative Review"

_biomedicines, 2025, doi:10.3390/biomedicines13051016_

Round 1
Reviewer 1 Report
Comments and Suggestions for Authors
In this review, the topic of management of acquired hypothalamic obesity after childhood-onset craniopharyngioma. This review aims to give an overview of Craniopharyngioma, to describe the range of mechanisms underlying the clinical presentation of this condition, diagnosis, treatment and future therapeutic options. The Authors presented an updated review on current knowledge on the Craniopharyngioma.
The strong points of this review are: meticulous bibliographic research and clear and accurated analysis of the literature data about the Craniopharyngioma in children. The Review is written very clearly and simply. Moreover, a figure that schematically presents the pathogenesis of Craniopharyngioma could facilitate the reading for pediatricians.
To sum up, this review is very useful for the pediatric endocrinologist, and above all family pediatrician, presenting the state of the art on this topic.
Moreover, I want to underline the didactic value of this review.
Author Response
Rebuttal - biomedicines-3496917 - Management of acquired hypothalamic obesity after childhood-onset craniopharyngioma - Hermann L. Müller, Oldenburg, Germany
Dear Editor,
Thank you for giving us the opportunity to resubmit a revised draft of our manuscript “Management of acquired hypothalamic obesity after childhood-onset craniopharyngioma – a narrative review” for publication in biomedicines. We appreciate the time and effort that you and the reviewers dedicated to providing feedback on our manuscript and are grateful for the insightful comments on and valuable improvements to our paper. We have incorporated all of the remarks made by the reviewers. Those changes are highlighted (yellow) within the manuscript. Please see below, for a point-by-point response to the reviewers’ comments and concerns. All page numbers refer to the revised manuscript file with tracked changes.
Reviewer 1:
In this review, the topic of management of acquired hypothalamic obesity after childhood-onset craniopharyngioma. This review aims to give an overview of Craniopharyngioma, to describe the range of mechanisms underlying the clinical presentation of this condition, diagnosis, treatment and future therapeutic options. The Authors presented an updated review on current knowledge on the Craniopharyngioma.
The strong points of this review are: meticulous bibliographic research and clear and accurated analysis of the literature data about the Craniopharyngioma in children. The Review is written very clearly and simply. Moreover, a figure that schematically presents the pathogenesis of Craniopharyngioma could facilitate the reading for pediatricians.
I am grateful for the reviewer’s comment and have added the below depicted figure to the revised manuscript (reproduced form Müller et al., 2019, Nat Rev Dis Primer). We have applied for copyright permission at SPRINGER/Nature for the picture and the SPRINGER editorial office sent us a message that this use of this figure is free for the publishing author, who I am.
Figure 1: Features of adamantinomatous craniopharyngioma (ACP) and papillary craniopharyngioma (PCP). Reproduced from: Müller, H.L. et al. (2019) Craniopharyngioma. Nat. Rev. Dis. Primers with kind permission of Springer Nature.
To sum up, this review is very useful for the pediatric endocrinologist, and above all family pediatrician, presenting the state of the art on this topic.
Moreover, I want to underline the didactic value of this review.
I am grateful for the reviewer’s comments.

Reviewer 2 Report
Comments and Suggestions for Authors
See the attached file

English Proof reading
Author Response
Rebuttal - biomedicines-3496917
Management of acquired hypothalamic obesity after childhood-onset craniopharyngioma
Hermann L. Müller, Oldenburg, Germany
Dear Editor,
Thank you for giving us the opportunity to resubmit a revised draft of our manuscript “Management of acquired hypothalamic obesity after childhood-onset craniopharyngioma – a narrative review” for publication in biomedicines. We appreciate the time and effort that you and the reviewers dedicated to providing feedback on our manuscript and are grateful for the insightful comments on and valuable improvements to our paper. We have incorporated all of the remarks made by the reviewers. Those changes are highlighted (yellow) within the manuscript. Please see below, for a point-by-point response to the reviewers’ comments and concerns. All page numbers refer to the revised manuscript file with tracked changes.
Reviewer 2:
Reviewer Comments The manuscript entitled “Management of acquired hypothalamic obesity after childhoodonset craniopharyngioma” is a comprehensive review article about the rare tumor craniopharyngioma and childhood obesity acquired after management with coverage of the diagnostic criteria, treatment strategies, and long-term outcomes. Here are some comments that might help your manuscript improvement.
- Title
The title is clear, concise, and accurately reflects the content of the manuscript. It specifies the focus on "acquired hypothalamic obesity" and "childhood-onset craniopharyngioma," which are key topics. Consider adding a phrase specifying the type of the review (narrative; systematic...etc.).
I am grateful for this comment and suggestion, which has been realized in the revised manuscript: Management of acquired hypothalamic obesity after childhood-onset craniopharyngioma – a narrative review.
- Abstract Ø
Lines 9–30: The abstract provides a good summary of the manuscript, including background, key findings, and conclusions. However, authors should determine the type of the review by adding a phrase specifying that this review address the importance of hypothalamic obesity as a significant clinical challenge and mentions promising therapeutic approaches. It lacks specificity regarding the methods used for the review. For example, it does not mention whether this is a systematic review, narrative review, or meta-analysis.
I am grateful for the suggestion and agree with the comment. The following text was added to the abstract: A narrative review was performed after search of the MEDLINE/ PubMed, Embase, and Web of Science databases for initial identification of articles. The search terms craniopharyngioma and hypothalamic obesity were used.
Line 17: "Despite the availability of promising therapeutic approaches…no pharmaceutical treatment has been proven effective in randomized controlled trials" is vague and needs elaboration.
Elaboration of this statement is given in the abstract and in the review.
Revise English and rephrase the abstract to include methodological details, clarify key findings, and improve sentence structure.
The abstract has been rewritten to include methodological details and the sentence structure was improved by an English proofreading agency performed by the editorial office of MDPI (MDPI author service).
- Introduction Ø
Lines 36–45: It is too short; add some phrases to introduce to the reader the definition, prevalence, complications of this tumor, origin…etc.
I am grateful for the reviewer’s comment and agree with the suggestion to extend the introduction and include further information on definition, prevalence, complications and other characteristics. I have added the following: Craniopharyngiomas (CP) are rare embryonal tumors of low-grade histological malignancy, located in the sellar/para- and supra-sellar region of the skull base. The WHO 2021 classification of central nervous system tumors defines adamantinomatous and papillary CP for the first time as two distinct tumor entities. The adamantinomatous CP (ACP) is mainly diagnosed in children and adolescents (<18 years), whereas the papillary CP (PCP) is diagnosed almost exclusively in the adult age group. The median age at diagnosis of adamantinomatous CP ranges from 5 to 9 years and from 55 to 69 years for papillary CP. The diagnosis of childhood-onset (CO) CP is often made late – sometimes years after the initial appearance of symptoms – with clinical picture at time of diagnosis often dominated by non-specific manifestations of intracranial pressure. Among adult-onset (AO) CP the diagnosis is maid mainly on visual symptoms but also on longstanding endocrine insufficiencies. The incidence of CP is 0.5–2.0 new CP cases per million persons per year. [1][2]. Despite high overall survival rates (92%), the quality of life after CP is frequently reduced due to adverse late effects caused by morbid obesity [3-5]. It is well known that the grade of hypothalamic obesity in affected CP patients is associated with the degree and extent of hypothalamic lesions [6].
Add phrases about the global burden or prevalence of craniopharyngioma worldwide or the country where the research was conducted.
Further epidemiological data on CP were included in the Introduction section.
The transition from general information about CP to its impact on hypothalamic obesity is abrupt. More transition phrases are needed.
Thank you for the suggestion, which we realized in the introduction of the revised manuscript.
Recent references are lacking update and review literature that is more recent.
More recent references are cited in the text of the revised review.
Add recent references, smooth transitions, and English proofreading is needed for the whole manuscript.
We added more recent references. English proofreading was performed by MDPI author service.
Ensure consistency in terminology and abbreviations. Do it all over the manuscript.
I checked the manuscript for consistency in terminology and abbreviations.
- Materials and Methods
This section was not mentioned. The authors should determine the type of the review, data bases used, literature inclusion and exclusion criteria, date, key words,…….etc. This transparency would enhance the credibility of the review.
I am grateful for the reviewer’s comment and suggestion. I have added a section “Materials and methods” to the revised manuscript: A scoping review was performed after search of the MEDLINE / PubMed, Embase, and Web of Science databases for initial identification of articles. The search terms craniopharyngioma and hypothalamic obesity were used. English language papers published between 1995 and 2024 were included in the search.
- Literature review:
Page 3, Lines 46–83: Define technical terms like "arcuate nucleus" and "paraventricular nucleus" for readers unfamiliar with neuroanatomy.
We have explained characteristics of these nuclei and depicted anatomical location in Figure 2.
This review reads like a list of facts rather than a narrative. For example, the section on hypothalamic nuclei (Lines 58–61) lacks integration with the broader discussion of hypothalamic obesity.
The connection is pointed out in the closing sentences of the section: Neurons in the paraventricular nucleus, the ventromedial nucleus, dorsomedial nucleus, and the lateral hypothalamic area control food intake [12]. The median preoptic nucleus, subfornical organ, and organum vasculosum form the thirst center in the lamina terminalis [12].
Outdated References: Many citations are from before 2015. For instance, the reference to "de Vile et al. (1996)" (Line 190) is outdated and should be updated with more recent studies.
Some “older” references (like de Vile 1996, and Müller 2001) represent “hall mark” publications on this topic, which are still relevant in spite of their old publication age and these publications are part of many reviews in this field.
Important topics such as genetic predispositions and novel imaging techniques are underexplored.
A genetic predisposition for craniopharyngioma is not reported. MRI and CT (for detection of calcifications) are the gold standards for imaging diagnostics.
Lines 84–102: Comprehensive overview of diagnostic imaging techniques (MRI, CT) and treatment options (surgery, radiotherapy). The discussion of risk-adapted strategies is particularly valuable. Try to include a table summarizing the pros and cons of different imaging modalities, lightening more the role of proton beam therapy in pediatric patients.
I think the challenges and limitations of different imaging modalities are sufficiently described in the manuscript. MRI and CT for detection of calcifications (limited sellar scans sparing the lenses, without intravenous contrast agent) are recommended as state of the art of imaging. Proton beam therapy is a radio oncological technique and – as I think - sufficiently described in terms of pros and cons in the manuscript.
Lines 179–200: Although the complications of craniopharyngioma were addressed well, try to review potential interventions to mitigate long-term complications (e.g., cardiovascular risks).
We added the information and discussed potential interventions to mitigate long-term complications (e.g., cardiovascular risks).
Lines 240–314: discuss more about that pharmacological treatment commenting on mechanism of action, side effects particulary thyroid cancer.
We are grateful for the reviewer’s comment. We added: Early studies suggested that GLP-1 receptor agonists might increase the risk of thyroid cancer [Parks et al., 2010]. Now, a large-scale Scandinavian study has found no significant increase in thyroid cancer risk in those taking GLP-1 receptor agonists compared with those on other treatments for type 2 diabetes [Pasternak et al., 2024].
Lines 315–339: Benefits and risks of bariatric surgery are well reviewed; however try to include a table summarizing data on patient-reported outcomes (e.g., quality of life) pre- and post-surgery.
We are grateful for the reviewer’s suggestion. I have included a new Table 2 summarizing selected literature on craniopharyngioma and quality of life (published between 2019 and 2025) in the revised manuscript.
- Figures and Tables Ø
Figure 1 provides a useful anatomical overview, and Table 1 summarizes therapeutic interventions effectively. However,
Figure 1: The figure lacks labels for key structures, making it difficult to interpret without prior knowledge. Mention whether the figure was created specifically for this manuscript or adapted from another sources. (add citation if any the reference 77 is incomplete please revise).
We are grateful for the reviewer’s comment and suggestion. We have added a new Figure 1 to our revised manuscript, which helps to better understand the anatomical situation of the “previous Figure 1 (now Figure 2). Reference 77 as source of the Figure was completed: modified from the doctoral thesis of Julia Beckhaus, Doctoral Thesis at School IV - Medicine and Health Sciences, Carl von Ossietzky Universität Oldenburg, Oldenburg, Germany.
Figure 2: lacks clear legend and description of the source (add citation if any).
The MRI and CT imaging of the figure are imaging of two patients with craniopharyngioma recruited in KRANIOPHARYNGEOM 2007, who gave the permission to publish their anonymized imaging.
Table 1: While informative, the table is not uniformed, please remake, unifying the statistical analysis used, make it consistent, and add a clear key and abbreviation details.
I am grateful for the reviewer’s comment. However, based on the published data a unifying of the used statistical methods for evaluation is not possible, because the necessary data for such a unifying process are not available in the cited publications.
- Discussion Ø
The section is repetitive, especially regarding the ineffectiveness of lifestyle modifications. Recent studies on GLP-1 receptor agonists and Setmelanotide are mentioned but not critically analyzed particularly the complications and increased risk f thyroid cancer.
We changed our revised manuscript accordingly and especially mentioned the potentially increased risks for thyroid cancer.
Transition phrases are lacking, making it had for the reader to follow.
We tried to improve the revised manuscript by changing this.
- Conclusion:
Rephrase adding more sentence for future research, and clinical implication.
I have rewritten the conclusion section following the reviewer’s suggestions.
- References
Many references are outdated. For example, Lustig (2008) and Mason et al. (2002) should be supplemented with studies that are more recent.
We substituted the cited paper of Mason et al. (2002) by a more recent publication on this topic by Denzer et al. (2019).
A clear material and methods with range of years, inclusion and exclusion criteria will help to improve this.
A material and methods section (including the above-mentioned information) was newly added to the revised manuscript.
DOI links are missing for several references, please follow all journal guidelines and formatting.
We followed the reference format (Chicago MDPI ens), that was mentioned in the Instructions to Authors of MDPI and provided by ENDNOTE® for BIOMEDICINE.
Check for plagiarism: Not detected
The editorial office (and/or the MDPI author service) is in charge of this and has already performed a plagiarism check.
- English and Grammar
The manuscript contains numerous grammatical errors and improper phrasing making it hard to follow. Simplify and break into shorter sentences. Use transition phrases and proofread the whole manuscript with English Editor.
We ordered MDPI author service to perform an English language check especially with regard to the above-mentioned points. This check has been performed and the changes of the manuscript are shown in the tracked format of the resubmitted revised manuscript.

Reviewer 3 Report
Comments and Suggestions for Authors
General concept comments
Acquired hypothalamic obesity after craniopharyngioma presents a major clinical challenge due to severe metabolic dysregulation and severe complication. The author presents the management of acquired hypothalamic obesity after childhood-onset craniopharyngioma effectively, but certain sections could be more concise and focused. A clearer structure with more defined subsections would enhance readability. Moreover, this is a purely review-based manuscript, clarify the selection criteria for included studies (e.g., systematic review vs. narrative review).
Specific comments
Abstract:
The abstract effectively outlines the scope of the review, emphasizing the significance of therapeutic approaches and currently treatment to improve management of hypothalamic syndrome/obesity.
Introduction:
The introduction should to provide a strong rationale for exploring pathophysiology of Craniopharyngiomas and acquired hypothalamic obesity, a severe complication that can occur after surgical or radiation treatment for craniopharyngioma. This will bring to keys point of why/how for management of acquired hypothalamic obesity after childhood-onset craniopharyngioma.
-Line 61, Figure 1: The reference [77] needs to rearrange.
-Section 3: suggestion for the 3. Pathophysiology of acquired hypothalamic obesity and diagnosis which include the key mechanisms and genetically, childhood-onset CP (Line 134-137)
-Figure 2 needs to clarify and point out the important spot hypothalamic damage which it related to acquired hypothalamic obesity.
Line 138-178: Treatment….this should including in section of “6. Multidisciplinary Management Approach acquired hypothalamic obesity” Because several Multidisciplinary have been used to help manage acquired hypothalamic obesity including 6.1 Pharmacological treatment, 6.2 Bariatric surgery, 6.3 Lifestyle modification,….. as show in Table1.
Discussion:
The discussion is thorough current knowledge on acquired hypothalamic obesity. The section would benefit from a focused discussion on how management of acquired hypothalamic obesity after childhood-onset craniopharyngioma modulation could be used in effective management strategies.
Conclusion
The conclusion highlights the rational of acquired hypothalamic obesity, but should include a stronger effective management. A final statement on the translational relevance of not require a “one-size-fits-all” management would not strengthen the impact.
Validity of the findings:
The manuscript is well-supported by references and relevant the clinical managements. However, the presents’ structure should clarify for more specific of effective requirement a multifaceted and individualized management.
Author Response
Rebuttal - biomedicines-3496917
Management of acquired hypothalamic obesity after childhood-onset craniopharyngioma
Hermann L. Müller, Oldenburg, Germany
Dear Editor,
Thank you for giving me the opportunity to resubmit a revised draft of my manuscript “Management of acquired hypothalamic obesity after childhood-onset craniopharyngioma – a narrative review” for publication in biomedicines. We appreciate the time and effort that you and the reviewers dedicated to providing feedback on our manuscript and are grateful for the insightful comments on and valuable improvements to our paper. I have incorporated all of the remarks made by the reviewers. Those changes are highlighted (yellow) within the manuscript. Please see below, for a point-by-point response to the reviewers’ comments and concerns. All page numbers refer to the revised manuscript file with tracked changes.
Reviewer 3:
General concept comments
Acquired hypothalamic obesity after craniopharyngioma presents a major clinical challenge due to severe metabolic dysregulation and severe complication. The author presents the management of acquired hypothalamic obesity after childhood-onset craniopharyngioma effectively, but certain sections could be more concise and focused. A clearer structure with more defined subsections would enhance readability. Moreover, this is a purely review-based manuscript, clarify the selection criteria for included studies (e.g., systematic review vs. narrative review).
I added a special section on “Materials and Methods” explaining the methodology and preparing of our scoping review. Additionally, the text of the revised manuscript was rewritten in certain parts to improve readability.
Specific comments
Abstract:
The abstract effectively outlines the scope of the review, emphasizing the significance of therapeutic approaches and currently treatment to improve management of hypothalamic syndrome/obesity.
I am grateful for the reviewer’s comment.
Introduction:
The introduction should to provide a strong rationale for exploring pathophysiology of Craniopharyngiomas and acquired hypothalamic obesity, a severe complication that can occur after surgical or radiation treatment for craniopharyngioma. This will bring to keys point of why/how for management of acquired hypothalamic obesity after childhood-onset craniopharyngioma.
I have rewritten the introduction and hopefully thereby improved the revised manuscript.
-Line 61, Figure 1: The reference [77] needs to rearrange.
I have rearranged the mentioned reference in the revised manuscript.
-Section 3: suggestion for the 3. Pathophysiology of acquired hypothalamic obesity and diagnosis which include the key mechanisms and genetically, childhood-onset CP (Line 134-137)
I would prefer to stick with current subheadings and structure.
-Figure 2 needs to clarify and point out the important spot hypothalamic damage which it related to acquired hypothalamic obesity.
Figure 2 depicts MRI and CT imaging of craniopharyngioma patients before surgical intervention. Accordingly, spotting surgical hypothalamic damage is not possible in this imaging.
Line 138-178: Treatment….this should including in section of “6. Multidisciplinary Management Approach acquired hypothalamic obesity” Because several Multidisciplinary have been used to help manage acquired hypothalamic obesity including 6.1 Pharmacological treatment, 6.2 Bariatric surgery, 6.3 Lifestyle modification,….. as show in Table1.
Discussion:
I am grateful for the suggestion, but would rather stick to the current structure. Section 3 is focusing on diagnosis and treatment of patients with craniopharyngioma. Section 6 is focusing on treatment of patients with acquired hypothalamic obesity, which comprises a different patient cohort and different and specific therapeutic approaches.
The discussion is thorough current knowledge on acquired hypothalamic obesity. The section would benefit from a focused discussion on how management of acquired hypothalamic obesity after childhood-onset craniopharyngioma modulation could be used in effective management strategies.
We have included this topic in the new conclusion section, which has been rewritten.
Conclusion
The conclusion highlights the rational of acquired hypothalamic obesity, but should include a stronger effective management. A final statement on the translational relevance of not require a “one-size-fits-all” management would not strengthen the impact.
I have followed these suggestions when rewriting the conclusion section.
Validity of the findings:
The manuscript is well-supported by references and relevant the clinical managements. However, the presents’ structure should clarify for more specific of effective requirement a multifaceted and individualized management.
I am grateful for this suggestion. In rewriting considerable parts of the manuscript (Introduction, conclusion and others) the recommended improvements hopefully were realized.

Reviewer 4 Report
Comments and Suggestions for Authors
- The review article "Management of acquired hypothalamic obesity after childhood-onset craniopharyngioma" discusses childhood-onset craniopharyngioma (rare, benign tumour near the pituitary and hypothalamic region of the brain) and the reduced quality of life due to acquire hypothalamic obesity. Therapeutic approaches to hypothalamic obesity include – surgery/radiotherapy to minimize craniopharyngioma-related hypothalamic damage, bariatric surgery and pharmacological agents – neuropeptides, antidiabetic medications and central stimulating agents. The article pointed out that despite current therapeutic approaches, there is a need for personalized treatment options. The single-author article has novel insight and can provide critical information for managing CP-linked childhood obesity. However, the draft is poorly conceived and not ready yet. It needs to be revised in several parts of the article. A few are suggested below.
- What is the methodology used in preparing this narrative review?
- The introduction (L37-45) is a crucial part of any narrative review. A more detailed introduction, including data on the prevalence, causality, genetics, and aetiology of acquired hypothalamic obesity, will engage the reader and set the stage for the review. It's important to identify the knowledge gaps and explain why this narrative review was necessary.
- Repeated statement from the abstract (lines 24-32) and conclusions part (lines 346-352)
- Repeated statements from the abstract (lines 10-16) and introduction part (lines 39-45)
- 1 shows the missing link between the Brain image and its detailed blow-up.
- Line 61 should write a "graphical software (biorender)" instead of biorender.com.
- 2 What is its source? Is this the original figure from a CP case? If yes, provide the details of the case in the figure legend. Ethical information should be provided.
- Lines 213-234 of the section read "odds", as these are described as bullet points, which can otherwise be placed as running text in a paragraph.
- L245-L250: Possible mechanisms involved in hypothalamic obesity can be separated into sections and elaborated on. This is crucial for understanding disease pathophysiology and might help improve clinical management, thus giving more structure to the review. Author is suggested to include deep brain (L241) stimulation (Table 1; ref 93) in the text with elaboration about the potential to improve quality of life in acquired hypothalamic obesity ( doi: 10.1016/j.heliyon.2023.e14411).
- Conclusions are the takeaway message for the reader from the author's side. At present, the section is repeated the narrative of the previous section without providing any novel information that addresses the proposed hypothesis conceptualized by the author at the beginning.
- The section should be a stand-alone and include a concluding statement rather than a question. Why problems is being cited (one size fits all) in the conclusion? The is no conclusive information on the treatment algorithm.
see before
Author Response
Rebuttal - biomedicines-3496917
Management of acquired hypothalamic obesity after childhood-onset craniopharyngioma
Hermann L. Müller, Oldenburg, Germany
Dear Editor,
Thank you for giving me the opportunity to resubmit a revised draft of my manuscript “Management of acquired hypothalamic obesity after childhood-onset craniopharyngioma – a narrative review” for publication in biomedicines. We appreciate the time and effort that you and the reviewers dedicated to providing feedback on our manuscript and are grateful for the insightful comments on and valuable improvements to our paper. I have incorporated all of the remarks made by the reviewers. Those changes are highlighted (yellow) within the manuscript. Please see below, for a point-by-point response to the reviewers’ comments and concerns. All page numbers refer to the revised manuscript file with tracked changes.
Reviewer 3:
- The review article "Management of acquired hypothalamic obesity after childhood-onset craniopharyngioma" discusses childhood-onset craniopharyngioma (rare, benign tumour near the pituitary and hypothalamic region of the brain) and the reduced quality of life due to acquire hypothalamic obesity. Therapeutic approaches to hypothalamic obesity include – surgery/radiotherapy to minimize craniopharyngioma-related hypothalamic damage, bariatric surgery and pharmacological agents – neuropeptides, antidiabetic medications and central stimulating agents. The article pointed out that despite current therapeutic approaches, there is a need for personalized treatment options. The single-author article has novel insight and can provide critical information for managing CP-linked childhood obesity. However, the draft is poorly conceived and not ready yet. It needs to be revised in several parts of the article. A few are suggested below.
- What is the methodology used in preparing this narrative review?
I added a special section on “Materials and Methods” explaining the methodology and preparing of our scoping review.
- The introduction (L37-45) is a crucial part of any narrative review. A more detailed introduction, including data on the prevalence, causality, genetics, and aetiology of acquired hypothalamic obesity, will engage the reader and set the stage for the review. It's important to identify the knowledge gaps and explain why this narrative review was necessary.
I have rewritten the introduction in the revised manuscript and furthermore added a new Figure 1. I think both changes and amendments reflect the reviewer’s suggestions.
- Repeated statement from the abstract (lines 24-32) and conclusions part (lines 346-352)
We have rewritten the conclusion section in the revised version of our manuscript.
- Repeated statements from the abstract (lines 10-16) and introduction part (lines 39-45)
I have changed this. The introduction was rewritten in the revised manuscript. There are no more repeated statements.
- 1 shows the missing link between the Brain image and its detailed blow-up.
- Line 61 should write a "graphical software (biorender)" instead of biorender.com.
We changed this in the revised manuscript.
- 2 What is its source? Is this the original figure from a CP case? If yes, provide the details of the case in the figure legend. Ethical information should be provided.
MRI and CT are diagnostic imaging of two patients with childhood-onset craniopharyngioma, who were recruited in KRANIOPHARYNGEOM 2007. Both patients gave their permission to publish their anonymized imaging slides.
- Lines 213-234 of the section read "odds", as these are described as bullet points, which can otherwise be placed as running text in a paragraph.
The bullets were inserted to mark a list of the domains, which are described in these sentence just before: Van Santen et al. have described five clinical domains for the diagnosis of hypothalamic syndrome: eating disorders, behavioral disorders, sleep disorders, temperature regulation disorders, and endocrine dysfunction [46, 47].
- L245-L250: Possible mechanisms involved in hypothalamic obesity can be separated into sections and elaborated on. This is crucial for understanding disease pathophysiology and might help improve clinical management, thus giving more structure to the review.
I have tried to give the section more structure.
- Author is suggested to include deep brain (L241) stimulation (Table 1; ref 93) in the text with elaboration about the potential to improve quality of life in acquired hypothalamic obesity ( doi: 10.1016/j.heliyon.2023.e14411).
We are grateful for the reviewer’s suggestion and have discussed the referenced paper on DBS in our revised manuscript.
- Conclusions are the takeaway message for the reader from the author's side. At present, the section is repeated the narrative of the previous section without providing any novel information that addresses the proposed hypothesis conceptualized by the author at the beginning.
I have rewritten the conclusion section in the revised version of our manuscript.
- The section should be a stand-alone and include a concluding statement rather than a question. Why problems is being cited (one size fits all) in the conclusion? The is no conclusive information on the treatment algorithm.
I have rewritten the conclusion section in the revised version of our manuscript.

Round 2
Reviewer 2 Report
Comments and Suggestions for Authors
Thank you for submitting the revised version of your manuscript.
Ø Title & abstract: The addition of specifying the type of review ("narrative review") has improved the clarity of the title.
The abstract has been significantly enhanced by including methodological details and improving sentence structure. Incorporating the phrase specifying the type of review and the search databases used adds transparency to the process. The elaboration on therapeutic approaches and the English proofreading improved the manuscript greatly. Introduction :
v The extension of the introduction with additional information on the definition, prevalence, complications, and other characteristics of craniopharyngioma is appreciated. The inclusion of recent epidemiological data and smoother transitions improves the overall flow and context setting.
v references are updated
Ø Materials and Methods:
The newly added section detailing the databases used, literature inclusion/exclusion criteria, and date ranges provides essential transparency and enhances the credibility of the review.
Ø Literature Review has greatly improved. The discussion on the risk-adapted strategies and imaging modalities is comprehensive. The inclusion of a new table (Table 2) summarizing patient-reported outcomes pre- and post-surgery is a valuable addition.
Addressing potential interventions for decreasing long-term complications and updating the discussion on pharmacological treatments and their side effects (e.g., thyroid cancer risks)
Ø Figures and Tables
Greatly improved
While unifying statistical analyses in Table 1 remains challenging due to the cited publications' limitations, the efforts to ensure clarity and consistency are evident.
The revised discussion section, particularly the critical analysis of GLP-1 receptor agonists and Setmelanotide, addresses previous concerns effectively. The transition phrases and reduced repetitiveness make this section more coherent and reader-friendly.
Finally, the revised version of the manuscript has been greatly improved and can be accepted for publication.
Reviewer 3 Report
Comments and Suggestions for Authors
After major revision this review is benefit for management acquired hypothalamic obesity. The authors provided comprehensive answers to all questions and incorporated my suggestions. Thank you for these efforts.
Author Response
ReRebuttal - biomedicines-3496917 04 08 2025
Management of acquired hypothalamic obesity after childhood-onset craniopharyngioma
Hermann L. Müller, Oldenburg, Germany
Dear Editor,
Thank you for giving me the opportunity to resubmit a revised draft of my manuscript “Management of acquired hypothalamic obesity after childhood-onset craniopharyngioma – a narrative review” for publication in biomedicines. We appreciate the time and effort that you and the reviewers dedicated to providing feedback on our manuscript and are grateful for the insightful comments on and valuable improvements to our paper.
Reviewer 3
After major revision this review is benefit for management acquired hypothalamic obesity. The authors provided comprehensive answers to all questions and incorporated my suggestions. Thank you for these efforts.
We are grateful for the reviewer’s comment.

Reviewer 4 Report
Comments and Suggestions for Authors
The revised draft, “Management of acquired hypothalamic obesity after childhood-onset craniopharyngioma—a narrative review,” improved significantly in readability and understanding. A few minor issues are listed below.
Line 37-39 The abstract conclusion must include both outcome and future work.
The introduction section needs to be read carefully, where sentences are written without a stop (line 52).
Line 62: The text insertion of Figure 1 is inappropriate with the statement. This is because the text should directly relate to the figure, providing a clear explanation of the data presented.
Line 71, the section can be merged without the title “Material and method”.
The introduction section should end with a clear statement about the rationale and purpose of this narrative review.
Comments on the Quality of English Languagesee before
Author Response
ReRebuttal - biomedicines-3496917 04 08 2025
Management of acquired hypothalamic obesity after childhood-onset craniopharyngioma
Hermann L. Müller, Oldenburg, Germany
Dear Editor,
Thank you for giving me the opportunity to resubmit a revised draft of my manuscript “Management of acquired hypothalamic obesity after childhood-onset craniopharyngioma – a narrative review” for publication in biomedicines. We appreciate the time and effort that you and the reviewers dedicated to providing feedback on our manuscript and are grateful for the insightful comments on and valuable improvements to our paper. I have incorporated all the remarks made by the Reviewer 3. Those changes are highlighted (green) within the manuscript. Please see below, for a point-by-point response to the reviewers’ comments and concerns. All page numbers refer to the revised manuscript file with tracked changes.
Reviewer 4:
The revised draft, “Management of acquired hypothalamic obesity after childhood-onset craniopharyngioma—a narrative review,” improved significantly in readability and understanding. A few minor issues are listed below.
We are grateful for the reviewer’s comment.
Line 37-39 The abstract conclusion must include both outcome and future work.
We are grateful for the reviewer’s comment and have included statements in the abstract on treatment and future work.
The introduction section needs to be read carefully, where sentences are written without a stop (line 52).
We are grateful for the reviewer’s comment and have inserted the missing stop.
Line 62: The text insertion of Figure 1 is inappropriate with the statement. This is because the text should directly relate to the figure, providing a clear explanation of the data presented.
We are grateful for the comment. We want to apologize for the mistake. The text insertion of Figure 1 is now placed at a more appropriate place in the revised manuscript.
Line 71, the section can be merged without the title “Material and method”.
We have merged the section ”Material and methods” with the text of the introduction as suggested by the reviewer. Accordingly, we have also renumbered the following headings.
The introduction section should end with a clear statement about the rationale and purpose of this narrative review.
We are grateful for the comment and have inserted the following sentence: This narrative review aims to present an overview over the current knowledge about hypothalamic obesity and its treatment, as well as future directions of research in this area.
